# Novel Multicomponent Digital Care Assistant and Support Program for People After Stroke or Transient Ischaemic Attack: A Pilot Feasibility Study

**DOI:** 10.3390/s24227253

**Published:** 2024-11-13

**Authors:** Liam P. Allan, David Silvera-Tawil, Jan Cameron, Jane Li, Marlien Varnfield, Vanessa Smallbon, Julia Bomke, Muideen T. Olaiya, Natasha A. Lannin, Dominique A. Cadilhac

**Affiliations:** 1Stroke and Ageing Research, Department of Medicine, School of Clinical Sciences at Monash Health, Victorian Heart Institute, Monash University, Clayton, VIC 3168, Australia; liam.allan1@monash.edu (L.P.A.); jan.cameron@monash.edu (J.C.); muideen.olaiya@monash.edu (M.T.O.); 2Australian e-Health Research Centre, Commonwealth Scientific and Industrial Organisation (CSIRO), Herston, QLD 4006, Australia; david.silvera-tawil@csiro.au (D.S.-T.); jane.li@csiro.au (J.L.); marlien.varnfield@csiro.au (M.V.); vanessa.smallbon@csiro.au (V.S.); julia.bomke@csiro.au (J.B.); 3Australian Centre for Heart Health, Royal Melbourne Hospital, Parkville, VIC 3052, Australia; 4Department of Neuroscience, School of Translational Medicine, Monash University, Melbourne, VIC 3004, Australia; natasha.lannin@monash.edu; 5Alfred Health, Melbourne, VIC 3004, Australia; 6Stroke Theme, Florey Institute of Neuroscience and Mental Health, University of Melbourne, Melbourne, VIC 3010, Australia

**Keywords:** stroke, secondary prevention, mobile health, telemedicine, pilot projects

## Abstract

Evidence is increasing for digital health programs targeting the secondary prevention of stroke. We aimed to determine the feasibility of the novel Care Assistant and support Program for people after Stroke (CAPS) or transient ischaemic attack (TIA) by combining person-centred goal setting and risk-factor monitoring through a web-based clinician portal, SMS messages, a mobile application (app), and a wearable device. We conducted a 12-week mixed-methods, open-label feasibility study. Participants (6 months–3 years after stroke or TIA, access to the internet via a smartphone/tablet) were recruited via the Australian Stroke Clinical Registry. Participants set one or two secondary prevention goals with a researcher and provided access and training in technology use. Feasibility outcomes included recruitment, retention, usability, acceptability, and satisfaction. Secondary outcomes included goal attainment, health outcomes, and program costs. Following 600 invitations, 58 responded, 34/36 (94%) eligible participants commenced the program (one withdrawal; 97% retention), and 10 were interviewed. Participants (27% female, 33% TIA) generally rated the usability of the mobile application as ‘Good’ to ‘Excellent’ (System Usability Scale). Most (94%) agreed the program helped with engagement in health self-monitoring. Overall, 52 goals were set, predominantly regarding exercise (21/52), which were the most frequently achieved (9/21). At 12 weeks, participants reported significant improvements (*p* < 0.05) in self-efficacy (Cohen’s d = 0.40), cardiovascular health (d = 0.71), and the mental health domain of the PROMIS GH (d = 0.63). CAPS was acceptable, with good retention and engagement of participants. Evaluation of this program in a randomised controlled trial is warranted.

## 1. Introduction

Globally, stroke is a leading cause of death and disability [1]. Concerningly, those who survive their event have a substantial risk of recurrence, with a rate of 16.7% reported at 3 years [2]. Patients who experience a transient ischaemic attack (TIA) are also at risk of experiencing a stroke, with a 7.3% risk within 12 months of the index event reported [3]. Therefore, it is a priority to minimise the risk of future vascular events in people with stroke and TIA.

Secondary prevention strategies, including lifestyle behaviour changes (e.g., increased physical activity) and pharmacological interventions, may prevent approximately 80% of recurrent events [4]. Self-management is crucial to the success of these proven strategies; however, the uptake and maintenance of these strategies are often sub-optimal. For example, in a retrospective cohort study of 9817 patients registered with the Australian Stroke Clinical Registry (AuSCR), nearly one-third had discontinued using their secondary prevention medication within a year [5]. Similarly, in a randomised controlled trial (RCT) of community-dwelling people with stroke, <12% of participants were physically active after 12 months, and <5% met recommended targets for daily consumption of vegetables and salt [6].

The use of technology to augment the delivery of healthcare may be one method to improve the secondary prevention of stroke (i.e., digital health). Digital health can facilitate continuous remote monitoring by clinicians and support the delivery of personalised patient-centred care [7]. For clinicians, the timely availability of relevant patient data can be used to enhance decision-making, diagnosis speed, and healthcare delivery [8]. For example, programs delivered using digital health have been used effectively to reduce blood pressure, improve glycaemic control, and improve the identification and management of atrial fibrillation [9,10,11]. There is also evidence that digital health programs have been successfully trialled in people living with stroke for rehabilitation to improve functional and psychosocial outcomes [12]. Despite their increasing use, there is little evidence available for the use of digital health programs in the secondary prevention of stroke [13], and fit-for-purpose programs are required.

Co-design is essential to develop such digital programs, increasing the likelihood of uptake by people living with chronic conditions and the clinicians providing their care [14,15]. To address this gap, we co-designed the Care Assistant and support Program for people after Stroke or TIA (CAPS) with people living with stroke or TIA (n = 112) and clinicians (n = 54), with the goal of improving secondary prevention of stroke [16]. Developed from a collaboration between CSIRO and Monash University, CAPS combines design features from past successful digital health projects. These include a cloud-based mobile technology-enabled rehabilitation platform [17] and an interface for scheduling electronic messages aligned to person-centred recovery and secondary prevention goals [18]. The resultant program is a multicomponent digital health program with clinician-led components (including person-centred goal setting, web-based clinician portal), goal-aligned short message system (SMS) messages, and a consumer-facing mobile application (app) coupled with a synchronised wearable device (smartwatch or activity tracker) for health monitoring. Offered as virtual delivery of stroke prevention care, CAPS also allows clinicians to remotely monitor and respond to participant health data collected through the app and wearable device via the clinician portal. The next research stage was to test the clinical protocol and program components in a feasibility study. We aimed to determine the feasibility of CAPS among people living with stroke or TIA in terms of recruitment, retention, usability, acceptability, and satisfaction.

## 2. Materials and Methods

### 2.1. Study Design

We conducted an open-label, single-group pretest–posttest sequential mixed-methods feasibility study. Data were obtained from app usage logs, physiological parameter monitoring, and baseline and follow-up outcome assessments. A satisfaction survey was completed by participants, and a purposive sample also participated in semi-structured interviews at the completion of the program. All participant data were entered into the secure web-based Research Electronic Data Capture (REDCap) platform [19], which was hosted by the CSIRO.

### 2.2. Participants

This study comprised people living with stroke or TIA and clinicians involved in the potential future delivery of the program. Patient participants were eligible if they (a) had a diagnosis of stroke or TIA in the previous 6 months to 3 years, (b) were aged ≥ 18 years old, (c) were living in the community, (d) were proficient in spoken and written English, and (e) had access to a smartphone with internet access.

This paper is focused on the experience of patient participants involved in the study. The feedback from clinicians potentially involved in the future delivery of the program will be reported in a separate publication.

### 2.3. Recruitment

Participants were recruited through the AuSCR, a national clinical registry designed to improve the quality of acute stroke care [20]. A mailout was conducted to 600 registrants of the AuSCR residing in the Australian states of Queensland, South Australia, Tasmania, and Victoria. Only AuSCR registrants who indicated in the routine follow-up at 90–180 days that they were willing to participate in future research were approached. Study invitation packs were distributed by the AuSCR between September 2022 and November 2022 and contained an explanatory statement, consent form, pre-enrolment survey, and pre-paid return envelope. Invitation packs were sent to people who were known to be alive (death records are updated each year by AuSCR) and who lived in the community within the designated Australian states.

The AuSCR registrants who returned a signed consent form and pre-enrolment survey were contacted by program coordinators (J.L., L.P.A.) via a telephone call within 1–2 weeks to confirm their eligibility, availability for commencement procedures, and preference for mode of completing surveys. If participants did not own their own health monitoring equipment, wearable devices and/or blood pressure monitors were posted to participants for use during their involvement in this study.

### 2.4. Baseline Data and Participant Onboarding to the Program

Participants completed baseline surveys electronically via a link to REDCap, on paper surveys distributed via post, or by a telephone call with a program coordinator, depending on their preferences. Once baseline surveys were completed, participants were contacted via telephone within one week to set one to two secondary prevention person-centred goals with trained program coordinators. Goals that were important to the participants were developed using a standardised template and procedures adapted from a phase III randomised controlled trial in progress [21,22]. Goals were developed to be specific, measurable, achievable, realistic, and timebound (SMART) [23], and participants received a copy of them via email. During this phone call, participants were trained to use the technology (mobile app and wearable device) before commencing this study. Participants were encouraged to email and call the program coordinators who onboarded them for support during this study if they had any technical issues.

### 2.5. Program Components

The co-designed program combined multiple technological components (Figure 1), as follows:
A remotely accessible web-based clinician portal, where participant profiles were created, secondary prevention SMART goal statements were uploaded, personalised health measurements were selected, SMS messages aligned to personalised goals were activated, and alert thresholds were set by program coordinators. Education links were uploaded to the app through the portal and personalised to individual participants by the program coordinator. Data from the app and the wearable device were automatically uploaded to the portal for clinician monitoring, with alerts activated if the data collected were above or below thresholds (e.g., blood pressure measurements). Alerts were flagged in the portal for review, with measurements for blood pressure, mood, and distress activating email alerts to the program coordinators to facilitate a rapid response. Alerts also triggered corresponding automated SMS text messages to participants.Motivational, goal-aligned SMS text messages were sent at a rate of two messages per week, with a message related to each SMART goal. Some messages contained education hyperlinks to various clinician-endorsed sources. The messages could be turned on/off within the app settings by the participant.A mobile app (available on iOS and Android devices) where participants completed a health check-in daily, with measurements tailored to each participant according to their baseline profile and specific goals identified (e.g., activity, alcohol intake, blood glucose, blood pressure, body weight, distress level, fatigue level, mood level, smoking, social connection, social support). When no check-in was completed for 5 days, an SMS message reminder was sent automatically. Participants could use the app’s health journal feature to type and audio-record notes and take and record photos. Within the health journal, medication reminders could also be entered (and toggled on and off) to notify participants daily on their phones and wearable devices at the selected time. Participants were able to review their data within the app, including all health check-in data, their goals, and their notes and meal photos. Web links to resources related to stroke symptoms, information, and self-management were also provided within the app.A Bluetooth synchronised smartwatch or fitness tracker (Apple Watch Series 6, Fitbit Sense 2, Fitbit Charge 4), passively collected participant data from the wearable before being automatically uploaded to the portal and app. These data included step count, heart rate, and sleep.

### 2.6. Outcome Assessments

Participants were notified via email to complete a 6-week outcome assessment and then a final 12-week outcome assessment. Similar to the baseline, the options for completing these assessments were as follows: (1) directly via a link to the online purposefully designed REDCap database, (2) returning a paper survey in the post, or (3) via a telephone call, depending on their preferences. A participant satisfaction survey was also sent with the final outcome assessments. Once completed, program coordinators contacted all participants via a telephone call to assess goal attainment, complete missing survey responses, collect information on adverse events and willingness to participate in interviews and any future research, organise the return of wearable devices/blood pressure monitors, when necessary, and remove access to the mobile app.

A purposively selected sample of participants (n = 10) based on demographics (age, sex, stroke or TIA) and level of engagement with the technology (e.g., received alerts for health measurements) were invited to participate in a further one-on-one interview to explore their perceptions of the program. Participants were asked open-ended questions to understand their experience of onboarding and using the technology, engagement with the program, level of support from program coordinators, perceived benefits, and the method and burden of completing surveys. Individual interviews were conducted virtually using the Webex video conferencing system between March and May 2023. Two program coordinators (J.L., L.P.A.) provided support to participants downloading and connecting to the Webex platform prior to each interview.

These one-on-one interviews lasted approximately 45 min each. Interviews were facilitated by an independent, experienced qualitative researcher (J.B.) who had not been involved in this study and a study investigator (J.C., or D.S.-T.) who had been involved in the program delivery with participants. Interviews were audio-recorded and professionally transcribed verbatim.

### 2.7. Outcomes

#### 2.7.1. Primary Outcomes

The primary outcomes of this study were feasibility measures, which were assessed as recruitment, retention, and usability and acceptability.

The feasibility of recruitment was determined by the proportion of responders to the mailout, eligibility of participants, and demographics of respondents compared to non-respondents. Retention was determined by withdrawals pre- and post-commencement of this study and completion of outcome assessments.

Usability was determined by engagement with the 12-week program and mobile app via data collected from app usage logs (frequency of daily health check-ins, number of text/audio notes recorded, number of participants who set up medication reminders, usage and synchronisation of sensor data from a wearable device, engagement with educational support, number of participants who turned off SMS messages in the app, and number of SMS hyperlinks accessed). Participants also completed the system usability scale, which is a rapid method developed to evaluate the subjective usability of technological systems [24].

Acceptability and satisfaction were determined using quantitative and qualitative data. Participant perceptions of the program were assessed using a survey (incorporating a rating scale), in which questions were based on the diffusion of innovation theory [25], to understand the compatibility of the program with their needs and daily routines, their empowerment experience, and their observability and satisfaction with using CAPS. Qualitative end-user feedback provided in the one-on-one interviews was used to further assess the adoption of the program components.

#### 2.7.2. Secondary Outcomes

Secondary outcomes included the attainment of goals and changes in health survey outcomes regarding psychological and physical health and wellbeing. Goal attainment was assessed using the standardised Goal Attainment Scaling-Light (GAS-Light) approach [26] Goals were rated on a 5-point scale, ranging from −2 to +2, and adjusted to reflect baseline function (−1, some function; −2, no function). A score of 0 equated to an expected level of attainment. Positive scores indicated somewhat better than expected (i.e., +1 score) or much better than expected level of goal attainment (i.e., +2 score). Negative scores indicated partial attainment (i.e., −1 score), no change from baseline (i.e., −1 score if some function at baseline, −2 score if no function at baseline), and worse than baseline (i.e., −2 score).

Psychological health was measured using the Depression Anxiety Stress Scale 21-item questionnaire (DASS-21), with higher scores indicating greater levels of depression, anxiety, or stress [27]. Social support and connection were measured using the Duke Social Support Index (DSSI), where a higher total score indicates better perceived social support [28]. Cardiovascular health was measured using the Life’s Simple 7 [29]. Scores were assigned to physical activity, diet, body mass index, smoking status, blood pressure, blood cholesterol, and blood glucose, and a greater overall score indicated better cardiovascular health. Due to limited data, a person-centred approach was used to assess and score physiological metrics (blood pressure, blood cholesterol, blood glucose) in lieu of the traditional measurement approach [29]. An ideal score indicated no disease diagnosis or treatment, an intermediate score indicated a diagnosis and treatment, and a poor score indicated a diagnosis but no treatment [30,31]. Overall health and wellbeing were measured using the Patient-Reported Outcomes Measurement Information System Global Health Scale (PROMIS GH), with greater overall scores indicating better perceived health and wellbeing [32,33]. Self-efficacy was measured using the Self-Efficacy for Managing Chronic Disease 6-item Scale (SEMCD6), where greater total scores indicate higher self-efficacy [34]. Diet quality was assessed using the University of Leeds Short-Form Food Frequency Questionnaire (SFFFQ), where greater total scores indicate higher diet quality [35].

#### 2.7.3. Program Delivery Costs

To estimate the cost per participant, we estimated the costs to deliver the program over 12 weeks. This included collecting information on the maximum and minimum time spent by program coordinators to onboard and offboard participants, their monitoring of participant data in the clinical portal, the number of responses to triggered data alerts, and the amount of technological or study support needed. The cost of sending SMS messages and the price of the off-the-shelf wearable devices were also recorded. The unit prices for these data were obtained from Australian sources and reported in 2023 Australian dollars (AUD) when these were not directly obtained as unit prices.

### 2.8. Sample Size

A formal sample size calculation was not undertaken due to this being an open-label feasibility study. Instead, we sought to recruit 30–40 people living with stroke or TIA, consistent with other similar feasibility studies for this type of early phase research [36,37,38,39].

### 2.9. Analysis

Quantitative data: Participant characteristics were summarised as frequency and percentages (categorical variables) and median and quartiles (Q1, Q3) (continuous variables), as relevant. For primary outcomes, recruitment, retention, acceptability, and satisfaction were summarised as frequencies and percentages, while usability was summarised as counts and means. For secondary outcomes, the attainment of goals within categories was summarised as frequencies and percentages. The GAS-T method was also used to assess goal attainment [26]. Individual goal scores (+2 to −2) were combined to produce an overall score and transformed into T-scores, where a T score of > 50 indicated that the goal was often attained, and a T score of <50 indicated that a goal was often not attained. Other continuous secondary outcomes (i.e., DASS-21, DSSI, LS7, PROMIS GH, SEMCD-6, and SFFFQ) were summarised as means and standard deviations.

Changes in health outcomes from baseline to 6 weeks and 12 weeks were determined using a paired-sample t-test, with the effect size expressed using Cohen’s d. In exploratory analyses, we determined correlations between measures of depression, anxiety, and distress (DASS21) and social connection and interaction (DSSI) collected via daily app check-ins, with self-reported measures of mood, distress, and social connection and support collected at the 6- and 12-week surveys. When survey data were missing at one time point, the last observation was carried forward.

Program costs were presented in different cost categories (e.g., staffing time, technology purchase, technology maintenance) and as an overall average cost/participant. Qualitative data: An inductive thematic analysis of the qualitative data from the one-on-one interviews was completed by two researchers (L.P.A., J.L.) using NVivo data management software (Version 12, QRS International Pty Ltd., Burlington, MA, USA). Codes were established based on the core topics of technology engagement experience, perceived impact, perceptions of this study, and potential implementation pathways. Themes and subthemes were generated based on their frequency of being mentioned and their relationship to the topics. Coding was cross-checked and discussed between the two researchers prior to the second round of coding processes and writing of the results. Triangulation of the data was used to explore and synthesise a more complete understanding of participant perceptions of the overall program and components, comparing the concordance of the data collected from surveys with feedback from interviews [40]. Illustrative quotes are provided.

## 3. Results

### 3.1. Feasibility of Recruitment and Retention

Recruitment for this study began in September 2022, with the final participant commencing this study in February 2023. Final outcome assessments were completed in May 2023. Of the 600 invitations sent to AuSCR registrants, 58 (10%) people returned their pre-enrolment surveys. Of those who responded, twenty-two (38%) were deemed ineligible (i.e., eight could not be contacted, four did not own a smartphone, three had incompatible smartphones, two had cognitive or physical difficulties that prevented use of the app, two were not comfortable using technology, two were travelling internationally during the trial period, and one would not fully consent). Among the remaining 36 eligible candidates, 2 withdrew prior to the completion of the baseline outcome assessments due to personal circumstances, with 34 participants commencing this study (Figure 2). One participant was withdrawn from this study after commencement due to a change in diagnosis, i.e., not stroke/TIA. Non-responders and ineligible participants were slightly older, were more often women, had a stroke, and were living in Queensland. Overall, 33 participants completed this study (97% retention). All but one survey (LS7 at baseline) was fully completed at all three time points by all participants. Most surveys were completed directly via REDCap (n = 31), with the remaining two participants opting for post or telephone calls.

### 3.2. Participant Characteristics

Participants in this study were recruited at a median (Q1, Q3) of 1.6 years (63.8, 74.5) from their last stroke or TIA, with about one-third experiencing a TIA (Table 1). The cohort had a median age of 71 years (53.9, 78.7), with fewer female participants (n = 9, 27%) (Table 1). Most participants identified as Australian (24/33, 73%) and were married or had a partner (n = 23, 70%), with nearly half the cohort living in the state of Tasmania (15, 46%). The majority of participants reported having a history of hypertension (n = 24, 73%), and 30% had atrial fibrillation.

### 3.3. Usability, Acceptability, and Satisfaction

#### 3.3.1. Data Usage Logs

During the 12-week (84-day) study duration, the daily check-in page was ‘clicked’ a total of 3169 times within the app (91 times/participant), with the check-in completed 2529 times (73 times/participant) (Table 2). In the last week of their program, 21 participants (64%) were still completing the daily check-in. As compared to those who stopped, more women continued completing the daily check-in (continued: 33% women, 67% men vs. stopped: 17% women, 83% men), self-reported having high blood pressure (continued: 81% yes vs. stopped: 50% yes) and high blood cholesterol (continued: 48% yes, stopped: 25% yes), and more had a stroke (continued: 76% vs. stopped: 50%). Participants most frequently viewed their health data (164 times/participant), with an average of almost twice a day per participant over the study duration. The health journal features were less frequently used, particularly the audio notes (1.5 times/participant over 12 weeks). Medication reminders were entered and activated within the app almost three times per participant. The stroke information page was accessed a total of 80 times (2.4 times/participant) over 12 weeks.

Overall, 294 unique SMS messages were sent to participants a total of 1492 times, with 98 containing active hyperlinks (33%). Messages containing hyperlinks were sent a total of 447 times and were accessed by participants 118 times (26%). Only three participants opted to turn off the SMS messages, which was possible through the app settings.

Wearable devices were mailed to 26 participants, including six Apple Watches (Series 6), 13 Fitbit Charge 4 fitness trackers, and seven Fitbit Sense 2 watches. The use of the wearable devices decreased over the study duration from 30 participants initially to 16 at the conclusion of the program, and a trend was observed for a small reduction in the average step count across the cohort (Figure 3).

#### 3.3.2. System Usability Scale

Ratings for the usability of the CAPS app ranged from 35 to 100 on the SUS, with an average of 76.5 (SD 16.5). Most participants (n = 24, 72.7%) scored the app above 68, giving the app a Good or Excellent usability rating per the SUS.

#### 3.3.3. Usability and Acceptability of the CAPS App

Over 90% of survey responses indicated that the app was not difficult to use due to any pre-existing health conditions. Generally, participants did not need assistance using the app. Nearly all responders (91%) agreed that the time taken to enter the health measures in the daily check-in was appropriate and that the health measures collected were relevant (82%). Participants who were interviewed indicated that their interest was maintained to continue to enter the health measures over the study duration.


*‘The questions were clear; I think it was very well put together’. *
*[P1]* 

Interviewed participants noted that the check-in questions and response features were easy to use and follow, but most did not complete the check-in daily, recommending they be reduced to three to four times a week. Additionally, some of the interviewed participants found it difficult to summarise their mood over 24 h in one measurement ranging from happy to sad, as follows:


*‘[…] in a day given what’s going on in the world and in my life, there would not be a day where I don’t have something sad and something happy, but I couldn’t define it as one or the other for a 24-h period, so I was fairly consistent about not describing myself as totally happy or totally sad’.*
*[P2]* 

In interviews, participants stated they found it ‘interesting’ to review their data, such as blood pressure measurements, as follows:


*‘[…] it was interesting to, […] look at the graphs, look at my blood pressure, pulse rates […] I found that interesting’. *
*[P4]* 

Less than half of the survey respondents (48%) felt the health journal feature within the app was useful. Only two interviewed participants used the medication reminders, with the other eight using pre-existing methods (e.g., Webster pack). Both participants found the reminders easy to activate and useful, and the notifications useful to ‘rejog their memory’ to take their medication.

While the interviewed participants accessed the education links within the app, some mentioned they were already aware of the information. They felt it would have been more useful in the early stages of recovery, given they were no longer in recovery from their stroke and had no ongoing deficits.


*‘[…] Fully recovered, only a slight stroke, […] did not use stroke info […] because I’d already read, and [I’m] educated, [could] probably work it out for myself’.*
*[P6]* 

In the satisfaction survey, five participants indicated that they were alerted to seek medical attention because their health measures entered in the app were outside the normal range. Four of these participants felt the advice was appropriate, with two acting on the advice to seek medical care.

#### 3.3.4. Wearable Device

The majority of survey respondents (85%) found the wearable devices convenient and easy to use, particularly for tracking and monitoring their health, with participants interviewed highlighting it made them feel ‘better informed’. Most respondents (76%) found the information collected and presented by the wearable was accurate. However, two interviewed participants felt the data collected by the wearable were already available in the wearable app, and they were unsure of the usefulness of duplicating that within the CAPS app. Thirteen participants requested to keep their wearable devices after the conclusion of this study so they could continue using them to monitor their health.

#### 3.3.5. Goal-Setting and SMS Messages

Overall, 70% of survey respondents were satisfied with the secondary prevention goals set with the program coordinators. Some of those who were interviewed reported having some difficulty in selecting goals due to feeling that their health was already well-managed due to the length of time since their stroke/TIA and that the level of specificity of the goals made it difficult. However, others noted that the goals were particularly helpful in improving their health. When queried about the most appropriate messaging modality (SMS vs. email messages) to support goal attainment, participants who were interviewed had a strong preference for SMS messages due to these being ‘quick and easy’. From the survey, only 58% of participants reported that the SMS messages related to their goals fit their needs. However, when queried in interviews, some participants found the messages and the links they contained motivational, as follows:


*‘How did it make me more conscious? Well, I just knew that I should be eating better than I was and […] I got texts […] on my phone with links to blood pressure and salt reduction and I now look at packages and things to see and compare salt, so that’s been a really good thing for me’.*
*[P7]* 

One of the interviewed participants shared that they had opted to turn off the SMS messages in the app, with three others sharing the opinion that they would be more appropriate for those in the early recovery stage or whose health was not managed well.

#### 3.3.6. Satisfaction with Overall Program

Overall, 90% of survey respondents agreed that the program was easy to incorporate into their daily routines (Figure 4), with 76% indicating that it fit well into their lifestyle and how they like to manage their health. Just over half the participants (52%) reported that CAPS improved their stroke/TIA knowledge, with slightly more (64%) finding that it improved their knowledge of the self-management of risks (Figure 4). Most survey respondents (70%) supported the program as conducive to the secondary prevention of stroke, and almost all (94%) agreed that the program helped them engage with monitoring their health (Figure 4). In interviews, participants reported feeling ‘more confident’ and proactive in their recovery, finding CAPS helped them stay on track with managing their health and lifestyle.


*‘I think it would be very good because always at the back of my mind, my greatest fear is that I’ll have another stroke—and when you’re doing something like this it sort of makes me feel more confident, and that I’m doing something’. *
*[P7]* 

In particular, those who were interviewed indicated that the program helped them monitor their blood pressure, improve their physical activity, lose weight, adhere to their medication, and establish new routines (e.g., medication adherence).


*‘Staying on top of my blood pressure regularly, you know that’s—I mean I was just relying on the medication before, that it was doing its job, but to actually […] be taking the blood pressure’. *
*[P1]* 

From the survey, few participants (27%) felt the effects of using the program were apparent to others, but a large number (79%) would recommend the program to other people. Most participants (76%) would use the program to manage their health, if available, and would discuss the recorded data with their healthcare providers during consultations (85%). For example, one interview participant reported that their specialist liked their involvement in the program and recommended the continued use of the wearable.


*‘When I went to see my […] heart specialist […] he said keep doing [the program], said it’s good, and I showed him I had the Fitbit, he said […] you keep wearing it, it’s very good for you’. *
*[P8]* 

### 3.4. Secondary Outcomes

#### 3.4.1. Goal Attainment

Overall, 52 secondary prevention goals were set, with 21 participants setting 2 goals, 10 setting 1 goal, and 2 participants opting to not set any goals due to reporting very well-managed health. The majority of goals were for increasing exercise (21/52, 40%), losing weight (12/52, 32%), and controlling blood pressure (5/52, 10%) (Appendix A). Overall, based on raw GAS scores, the goals were most often partially achieved (29%), or there was no change from baseline (25%). However, nearly 20% of all goals were achieved, which was a lot more than expected; the achieved goals particularly involved exercise (4/21, 19%) and feeling less depressed and anxious (3/3, 100%). Only two participants had one goal each that they were farther from meeting at 12 weeks than at baseline; these pertained to exercise (1/21, 5%) and managing diabetes (1/2, 50%). Across the entire cohort, goal attainment using the mean GAS T scores was not achieved (mean 48.3, SD 13.5).

#### 3.4.2. Health Survey Outcomes

The use of CAPS was associated with a significant reduction in anxiety (DASS-21) by −1.58 (Cohen’s d = 0.36, *p* = 0.04) from baseline to 6 weeks; however, this was not sustained at 12 weeks. Overall, at 12 weeks, there were no changes in depression, anxiety, or stress (DASS-21), social support or interaction (DSSI), or dietary intake (SFFFQ) from baseline (Table 3). However, CAPS was associated with significant improvements in cardiovascular health (LS7), with a mean score increase from baseline to 12 weeks of 0.94 (d = 0.71) and in the domains of mental health (PROMIS GH) (4.65, d = 0.63) and self-efficacy (SEMCD-6) (0.48, d = 0.40) (Table 3).

#### 3.4.3. Correlations Between Survey Outcomes and Daily Check-In Measures

Moderate correlations were observed between mood and the DASS21 depression domain (*r* = 0.50), the social connection measure and the DSSI social interaction domain (*r* = 0.57), and the social support measure and the DSSI social interaction domain (*r* = 0.66) at 12 weeks (Table 4). Similarly, moderate correlations were also observed between the social support measure and the total DSSI score at 6 weeks (*r* = 0.55) and 12 weeks (*r* = 0.62). Strong correlations were observed between the social connection measure and the DSSI social connection domain (*r* = 0.74) and total DSSI score (*r* = 0.72) at 12 weeks (Table 4).

### 3.5. Preliminary Research Program Costs

Program coordinators took approximately one hour (minimum 45 min, maximum 3 h) to onboard each participant in the program (Table 5). Set-up tasks included developing secondary prevention goals with the participant, generating a participant profile in the clinical portal, and setting up and guiding the participant through using the wearable device and app. Some participants required a significantly longer period of time, particularly in setting up the wearable device and app when they were less technologically literate. Over the 12-week study duration, program coordinators spent a total of one hour per participant monitoring their data through the clinical portal, responding to clinical alerts, and assisting with technological issues. The dedicated software engineer spent approximately 1 h per week maintaining the technology.

The average overall cost of delivering CAPS was AUD 17,854 (minimum AUD 14,844, maximum AUD 20,865). The greatest cost incurred was the purchase of the devices (Table 5). The average program cost per participant over the 12-week study duration was AUD 541 (AUD 450–632).

## 4. Discussion

We report the feasibility of delivering a new, multicomponent digital program to support secondary stroke prevention by improving self-management and person-centred goal attainment in the community. We used a novel method of participant recruitment via a clinical quality registry, whereby 10% of the invited AuSCR registrants agreed to participate, achieving our desired sample size (≥30 participants) within 11 weeks of the first mailout. Through observing the data logs, we saw a high level of participant retention and engagement with the program components. Feedback was collected using an acceptability and satisfaction survey, which was supported by interviews with participants, and allowed us to evaluate the acceptability and appropriateness of the program. Participants found the different technological components easy to use and would use the program if available. Participants found the app to be highly usable, and the collected measures were considered appropriate. The wearables were easily integrated into their daily life and proved beneficial for tracking their health. They were generally satisfied with the goals that were set. However, they found the supporting SMS messages to be less useful. While secondary prevention goals were generally not attained, it was promising to see that most participants were able to improve their health behaviours and at least partially achieve these goals. Even with the small sample size, our program was associated with individual improvements in cardiovascular health, a domain of mental health, and self-efficacy. This evidence supports the potential use of CAPS as a program to empower patients to improve self-management, reduce their cardiovascular risk, and improve their overall wellbeing.

Another advantage of our study was the ability to demonstrate an entirely virtual model of care, in which we were able to conduct all processes remotely. Participants were successfully onboarded and familiarised with the technology, including those who were older and/or experiencing post-stroke impairments (cognitive, visual, and hand–motor function).

By offering a virtual program that is highly accessible, patients from remote or rural settings living with chronic disease are able to receive the same standard of care as their metropolitan counterparts, and access clinical expertise not available to them otherwise. Given that people living in remote or rural communities are generally at a greater risk of a recurrent event [41], providing access to these patients could have significant benefits in reducing rates of recurrence. However, several responders to the invitation mailout were ineligible because they did not own a compatible smartphone, which is a recurrent issue in digital health research, leading to inequities in receiving care [13]. Future digital health studies should explore various recruitment strategies and maximising accessibility (e.g., device provision and training), particularly for less digitally literate individuals.

While participants found the technology usable and easily integrated into daily life, there was a small trend of reduced usage by the later stages of this study, which is not uncommon for digital health programs. However, we observed much higher sustained usage of the technological components (64%) compared to the literature, with estimations of as few as 10–25% of recruited participants continuing to engage by the conclusion of other studies lasting between 1 and 12 weeks [42]. The reduction in usage may have been associated with the lack of feedback provided during this study regarding progress made by participants or the lack of scheduled clinical support [43]. To address the reduction in usage, the refined program, which will be tested in an RCT, has incorporated a standard clinical review during the program duration to encourage engagement.

CAPS is designed to support self-management primarily through clinician-led person-centred goal setting and subsequent supported self-monitoring. However, two participants did not set any goals when invited at baseline on the basis of having well-managed health. While the program coordinators who developed the goals with participants received training in person-centred goal setting, neither were clinicians experienced in health promotion or the care of people living with stroke or TIA. Consequently, they may have lacked the experience to encourage participants who felt their health was well-controlled to develop secondary prevention goals. Further investigation revealed that both participants who did not identify any goals were less physically active than recommended by clinical guidelines, with one also reporting hypercholesterolemia. Therefore, these participants could have been encouraged to develop goals for secondary prevention in these areas. As an outcome of this study, we will review procedures and the clinical protocol for the next stage of the evaluation of CAPS to maximise participants’ potential benefits from the program and ensure that all components are delivered as intended.

Although the majority of participants were satisfied with the secondary prevention goals, not all of them reported that the goals met their needs. When queried in interviews, participants reported few unmet needs that could have been improved through goal setting due to the amount of time since their index event. Subsequently, the goal-aligned SMS messages may have been less effective in motivating behaviour change and elicited mixed responses. These participants emphasised that goal setting and the resources provided during the program would have been more valuable in the early recovery stages following their stroke or TIA. In response to this feedback, the next phase of evaluating CAPS will target participants within the first 3–6 months of stroke or TIA.

While there was no comparator, this study provided evidence of associations with the use of CAPS and promising signals of health effects within subjects between baseline and 12 weeks. An average improvement of nearly one point in the LS7 demonstrated the potential use of CAPS to improve overall cardiovascular health. In previous research, it has been demonstrated that a one-point improvement in the LS7 is associated with a clinically meaningful 8% reduced risk of first-ever stroke [44]. Furthermore, improvements in self-efficacy were observed, which is associated with improved post-stroke recovery and the improved implementation and maintenance of behaviour change [45,46]. In the present study, we found moderate correlations between the measure of mood and the DASS-21 depression subscale, as well as the social connection measure and the DSSI social connection subdomain. Given this, these measures, which were entered rapidly through the app, could be used to more easily and frequently capture the wellbeing of people living with stroke rather than having them complete extensive surveys that entail a greater burden.

### Limitations

The time since stroke or TIA was a factor that influenced the engagement or perceived value of CAPS since participants were recruited at a median of 1.6 years from their last event. When registrants within the AuSCR confirm their willingness to participate in future research, it is generally 4–5 months from their hospital admission for stroke or TIA [47,48]. Given this circumstance, we were unable to evaluate the program among people living with stroke or TIA in the early recovery phase (i.e., in the first 3 months), when they may receive more benefit from goal setting, self-monitoring, and health promotion education. Some of the reduced usage of the app and wearable may have been attributable to participants being offboarded a few days prior to their final day in this study (80 days vs. 84 days) due to a scheduling error. Additionally, we did not collect any physical activity data at baseline, so what appeared to be reductions in physical activity from the early to later stages of this study may have been participants returning to baseline levels. Findings on health outcomes should be interpreted in the context of the limitations of the study design and need to be confirmed with more robust study designs, for example, an RCT.

## 5. Conclusions

In this study, we have successfully demonstrated that the co-designed program is feasible and acceptable and has the potential to benefit cardiovascular and psychosocial health. Further evidence for sustaining engagement with the technology, secondary prevention goal setting, and the maintenance of the program benefits beyond 12 weeks is required, given that people living with stroke need to consistently maintain these lifestyle behaviour changes in order to reduce their risk of recurrence. These data will inform the design of a phase II RCT to explore preliminary effectiveness, the most appropriate delivery format for this novel model of care, and the future study design, in addition to clinician perceptions and feedback from focus group/interviews, which will be presented in a complementary paper.

## Figures and Tables

**Figure 1 sensors-24-07253-f001:**
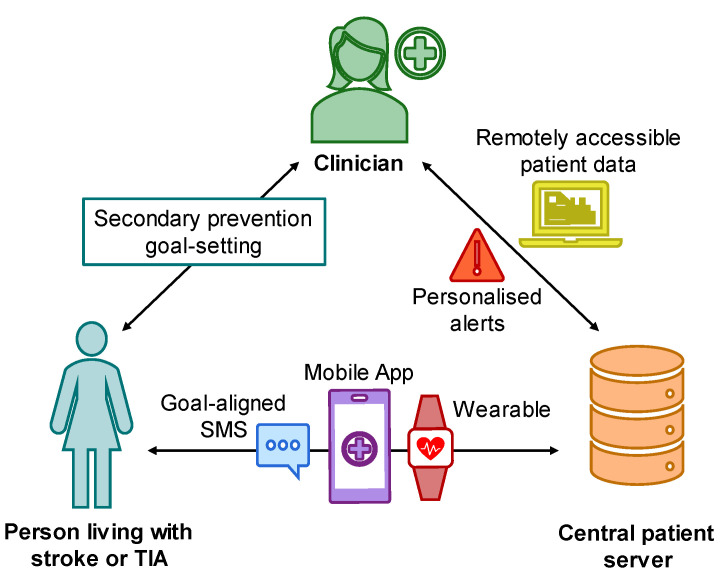
Schematic of the multicomponent program. SMS: short message service; TIA: transient ischaemic attack.

**Figure 2 sensors-24-07253-f002:**
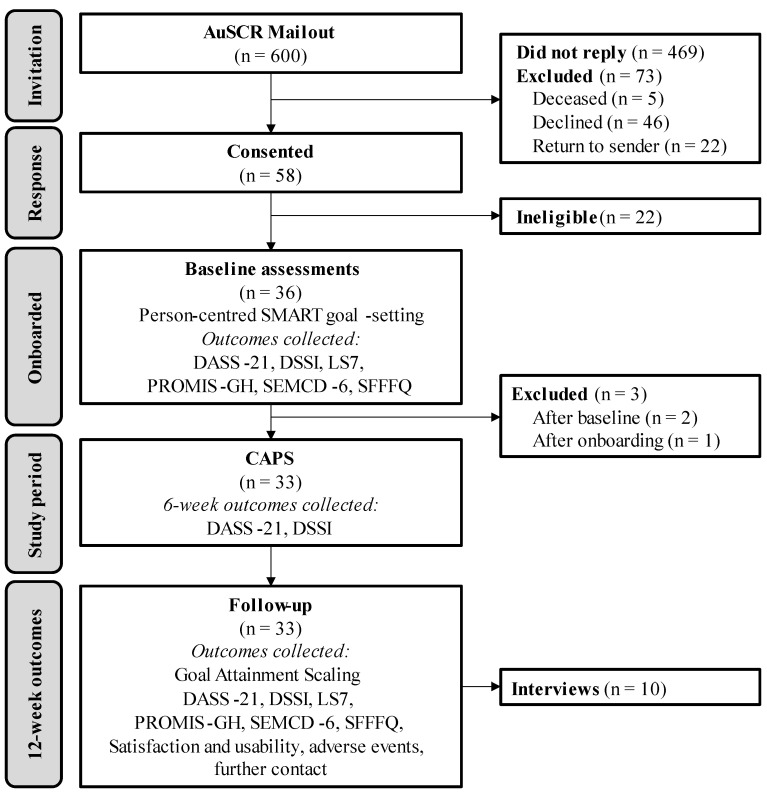
Consolidated Standards of Reporting Trials (CONSORT) flow of participant recruitment and outcome assessments. DASS-21: Depression Anxiety Stress Scale 21-item; DSSI: Duke Social Support Index; LS7: Life’s Simple 7; PROMIS GH: Patient-Reported Outcomes Measurement Information System Global Health; SD: standard deviation; SEMCD-6: Self-Efficacy for Managing Chronic Diseases 6-item Scale; SFFFQ: Short-Form Food Frequency Questionnaire.

**Figure 3 sensors-24-07253-f003:**
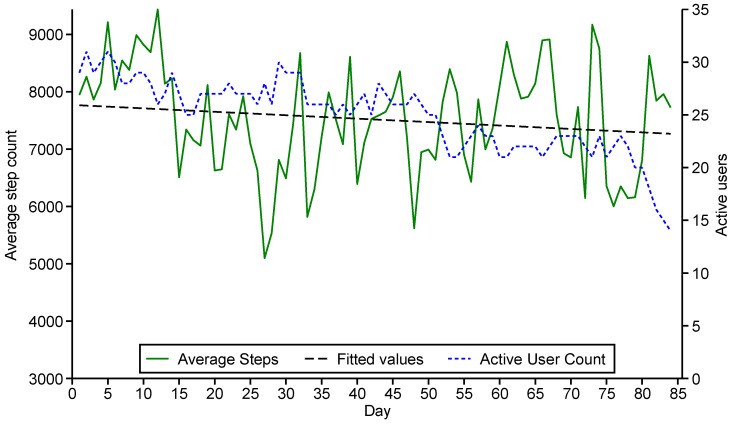
Number of active wearable device users and average steps per day over study period. Fitted line shows trend in reduced average steps over study period.

**Figure 4 sensors-24-07253-f004:**
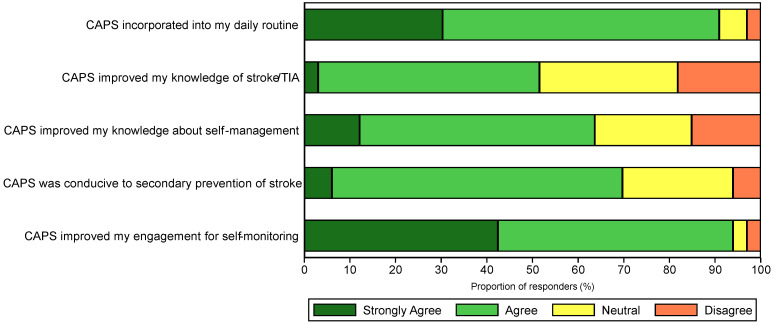
Perceived benefits and satisfaction with the overall program.

**Table 1 sensors-24-07253-t001:** Characteristics of study participants and those interviewed one-on-one.

Baseline Characteristics	Frequency, n (%)	
	Participants (N = 33)	Interviewed (N = 10)
**Demographics**		
Age in years, median (Q1, Q3)	70.5 (53.9, 78.7)	70.9 (68.3, 75.6)
Female	9 (27)	4 (40)
Australian	24 (73)	5 (50)
Married/with partner	23 (70)	6 (60)
Live in own home	28 (85)	9 (90)
Location within Australian states		
Queensland	4 (12)	2 (20)
South Australia	6 (18)	1 (10)
Tasmania	15 (46)	5 (50)
Victoria	8 (24)	2 (20)
Transient Ischaemic Attack (TIA)	11 (33)	3 (30)
Years since last stroke/TIA ^a^, median (Q1, Q3)	1.6 (1.3, 2.1)	2.0 (2.0, 2.1)
**Self-reported medical history**		
High blood pressure	23 (70)	8 (80)
High cholesterol	13 (39)	2 (20)
Diabetes mellitus	5 (15)	1 (10)
Atrial fibrillation	10 (30)	2 (20)
Speech or communication impairment		
Mild	4 (12)	1 (10)
Severe	1 (3)	-
Vision impairment		
Mild	5 (15)	3 (30)
Hand–motor impairment		
Mild	9 (27)	2 (20)
Moderate	2 (6)	2 (20)
Severe	2 (6)	-

^a^ At the time of recruitment. TIA: transient ischaemic attack; Q1: Quartile 1; Q3: Quartile 3.

**Table 2 sensors-24-07253-t002:** CAPS smartphone app usage counts by feature.

Event Name	Total Count	Average per Participant (N = 33)
**Daily Check-in**		
Check-in Started	3169	91.4
Check-in Completed	2529	72.6
**Review Your Health Data**	5719	164.3
**Health Journal**		
New Text Note	566	24.7
Review Text Note	299	15.8
Medication Reminder	71	2.9
New Audio Note	6	1.5
Review Audio Note	1	1.0
**Stroke Information**	80	3.0

**Table 3 sensors-24-07253-t003:** Differences in mean outcome measures at 12 weeks.

Outcome	Mean Outcome Measures (SD) N = 33		Mean Change T_0_ − T_2_ (SD)	95% CI	Effect Size
	Baseline (T_0_)	12 Weeks (T_2_)			
**DASS-21**					
Depression	6.37 (7.29)	5.76 (5.87)	−0.61 (4.49)	−2.20, 0.98	0.14
Anxiety	5.21 (6.59)	4.36 (5.56)	−0.85 (4.33)	−2.38, 0.69	0.20
Stress	7.88 (7.56)	7.58 (5.91)	−0.30 (5.03)	−2.09, 1.48	0.06
**DSSI**	27.18 (3.60)	27.15 (3.63)	−0.03 (2.72)	−1.00, 0.93	0.01
Social support	18.12 (2.70)	18 (2.70)	−0.12 (2.16)	−0.89, 0.65	0.06
Social interaction	9.06 (1.98)	9.15 (1.56)	0.09 (1.23)	−0.35, 0.53	0.07
**LS7 ^a^**	9.50 (0.29)	10.44 (0.31)	0.94 ** (0.23)	0.46, 1.41	0.71
**PROMIS GH**					
Mental Health	47.68 (9.64)	52.32 (9.95)	4.65 ** (7.33)	2.05, 7.24	0.63
Physical Health	48.73 (9.72)	51.27 (10.27)	2.55 (8.02)	−0.30, 5.39	0.32
**SEMCD-6**	7.33 (1.97)	7.82 (1.88)	0.48 * (1.19)	0.06, 0.90	0.40
**SFFFQ**	11.21 (1.47)	11.36 (1.52)	0.15 (1.68)	−0.44, 0.75	0.09

^a^ n = 32, * *p* < 0.05, ** *p* < 0.001. CI: confidence interval; DASS-21: Depression Anxiety Stress Scale 21-item; DSSI: Duke Social Support Index; LS7: Life’s Simple 7; PROMIS GH: Patient-Reported Outcomes Measurement Information System Global Health; SD: standard deviation; SEMCD-6: Self-Efficacy for Managing Chronic Diseases 6-item Scale; SFFFQ: Short-Form Food Frequency Questionnaire.

**Table 4 sensors-24-07253-t004:** Correlations between psychosocial health outcome surveys at 6 and 12 weeks with psychosocial daily check-in measures.

Survey		*r*	
		6 Weeks	12 Weeks
**DASS21**		**Mood**	
	*Depression*	−0.45	−0.50
	*Anxiety*	−0.01	−0.11
	*Stress*	0	0.01
		**Distress**	
	*Depression*	−0.33	−0.23
	*Anxiety*	−0.16	−0.27
	*Stress*	−0.33	−0.26
**DSSI**		**Social connection**	
	*Social connection*	0.44	0.74
	*Social interaction*	0.33	0.57
	*Total score*	0.49	0.72
		**Social support**	
	*Social connection*	0.41	0.47
	*Social interaction*	0.43	0.66
	*Total score*	0.55	0.62

*r*: Correlation coefficient. DASS21: Depression Anxiety Stress Scale 21-item; DSSI: Duke Social Support Index.

**Table 5 sensors-24-07253-t005:** CAPS delivery costs in the feasibility study.

	Cost (AUD)	Description of Unit Costs
**Purchase costs**		
Wearable devices	AUD 7493	Retail price of 7 Fitbit Sense 2 (AUD 449.95), 13 Fitbit Charge 4 (AUD 149.95), and 6 Apple Watch Series 6 (AUD 399)
Blood pressure monitors	AUD 495	Retail price of 5 BP monitors (AUD 99)
**Delivery costs**		
Program coordinator time	AUD 5744–AUD 11,765	Total minimum and maximum coordinator time, recorded in hours ^a^; labour time valued using the Monash University Research Professional Salary Schedule, including oncosts ^b^
Software engineer time	AUD 1061	Engineer time recorded in hours; labour time valued using the Monash University Research Professional Salary Schedule, including oncosts ^b^
SMS messages	AUD 52	AUD 0.035 per message
**Cost per participant** **(N = 33)**	AUD 450–AUD 632	Minimum and maximum cost per participant

AUD: Australian dollars; ^a^ minimum time: 1.45 h/participant, maximum time: 4 h/participant; ^b^ labour time was additionally costed at 1.25 to account for casual loading.

## Data Availability

Data are contained within the article.

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
