# Peer review of "Novel Multicomponent Digital Care Assistant and Support Program for People After Stroke or Transient Ischaemic Attack: A Pilot Feasibility Study"

_sensors, 2024, doi:10.3390/s24227253_

Round 1

Reviewer 1 Report

Comments and Suggestions for Authors

Sensors-3277731-peer-review comments:

The authors of this paper focus on using technologies, including wearable device and App, to improve secondary prevention of stroke. This article presents a novel Care Assistant and support Program (CAPS), a multicomponent digital health intervention designed to support secondary prevention in patients after stroke or transient ischemic attack (TIA). It can be referred as a model for the applications of wearable devices and APP. This study encourage me to do similar study to improve human being’s health with wearable devices and APP. This article is well-written, it can be accepted as current form. But from my opinion, in section 3.2.3, the comments from participants can be omitted for concise reason. However, the existence of these comments make this article vivid. The decision will be made by editorial board.  

Author Response

Reviewer 1

The authors of this paper focus on using technologies, including wearable device and App, to improve secondary prevention of stroke. This article presents a novel Care Assistant and support Program (CAPS), a multicomponent digital health intervention designed to support secondary prevention in patients after stroke or transient ischemic attack (TIA). It can be referred as a model for the applications of wearable devices and APP. This study encourage me to do similar study to improve human being’s health with wearable devices and APP. This article is well-written, it can be accepted as current form. But from my opinion, in section 3.2.3, the comments from participants can be omitted for concise reason. However, the existence of these comments make this article vivid. The decision will be made by editorial board. 

Response: We thank the reviewer for their constructive feedback. As suggested, we have made minor changes to make section 3.2.3 more concise, while still retaining some of the participant quotes that we felt were pertinent to highlight their feedback on the usability and acceptability of CAPS. The edited section is copied below (pages 11 & 12)

“The questions were clear; I think it was very well put together. I’m not just saying that to make you guys feel good, but it was well put together.” [P1]

In interviews, participants stated they highlighted the value of being able to review their data, findingfound it “interesting” to use the graphs to review historical their data, in-cluding for measurements such as blood pressure measurements.

While the interviewed participants did access the education links within the app, some mentioned they were already aware of the information. They felt it would have been more useful in the early stages of recovery, given they were no longer in recovery from their stroke and had no ongoing deficits., and that it would have been more useful in the early stages of recovery.

In the satisfaction survey, During the study, five participants indicated in the survey they were alerted to seek medical attention because their health measures entered in the app were from outside a normal range. out-of-threshold measurements entered into the app, Four four of these participants whom felt the advice was appropriate, with two acting on the advice to seeking out medical care as a result.

Reviewer 2 Report

Comments and Suggestions for Authors The article is exceptional in every respect, without any complaint. If there were any flaws in the submitted manuscript, I would clearly suggest it to the authors or the editor of the journal, without any problems. The submitted manuscript covers the topic of the journal in every respect. The content of the introduction provides enough quality information for the subject study, guided by current references of recent date, which are in the domain of research. The design of the article is in accordance with methodological standards that can enable the repetition of the same similar research. All methods were properly applied in accordance with statistical procedures, which enabled the correct interpretation of the results presented in the tables and figures. The discussion connected the results with the cause-and-effect relationships of the defined research problem. The conclusions are correct and clearly drawn in accordance with the obtained results.

A well chosen research problem. Very relevant in today's age. The target is group is well chosen. The results can serve very well in practice. 

Congratulations!

Author Response

Reviewer 2

The article is exceptional in every respect, without any complaint. If there were any flaws in the submitted manuscript, I would clearly suggest it to the authors or the editor of the journal, without any problems. The submitted manuscript covers the topic of the journal in every respect. The content of the introduction provides enough quality information for the subject study, guided by current references of recent date, which are in the domain of research. The design of the article is in accordance with methodological standards that can enable the repetition of the same similar research. All methods were properly applied in accordance with statistical procedures, which enabled the correct interpretation of the results presented in the tables and figures. The discussion connected the results with the cause-and-effect relationships of the defined research problem. The conclusions are correct and clearly drawn in accordance with the obtained results. A well chosen research problem. Very relevant in today's age. The target is group is well chosen. The results can serve very well in practice. Congratulations!

Response: We thank the reviewer for their time in reviewing our paper and the positive feedback received.

Reviewer 3 Report

Comments and Suggestions for Authors

The aim of the paper is to determine feasibility of the novel Care Assistant and support Program for people after Stroke (CAPS) or transient ischaemic attack (TIA), combining person-centred goal setting and risk-factor monitoring through a web-based clinician portal, SMS messages, a mobile application (app), and a wearable device. The main strength of this research lies in the improvement of self-monitoring of patients, prevention and monitoring of patients.

The manuscript is clear, relevant for the field and it is adequately structured.

The cited references are mostly recent and relevant  publications.

The manuscript sounds scientifically and it is appropriately  designed.       Prevention plan is thought out as goal settings which is person-centred and based on risk-factor monitoring. Risk-factor monitoring is very detailed and practical in the same time (web-based clinician portal, SMS messages, a mobile application (app), and a wearable device) .

Secondary prevention goals are set so that they are both expertly guided and that the needs of the patients are listened to (the study participants-patients set them together with the researchers).

 They provided access and training in technology use. Feasibility outcomes included recruitment, retention, usability, acceptability, and satisfaction. Secondary outcomes included goal attainment, health outcomes, and program costs.

Four figures and five tables are sufficiently informative, but also transparent and easy to understand.

The conclusions consistent with the evidence and arguments presented.

The data and analyses are presented statistically appropriately.

The English language appropriate and understandable.

Author Response

Reviewer 3

The aim of the paper is to determine feasibility of the novel Care Assistant and support Program for people after Stroke (CAPS) or transient ischaemic attack (TIA), combining person-centred goal setting and risk-factor monitoring through a web-based clinician portal, SMS messages, a mobile application (app), and a wearable device. The main strength of this research lies in the improvement of self-monitoring of patients, prevention and monitoring of patients. The manuscript is clear, relevant for the field and it is adequately structured. The cited references are mostly recent and relevant publications. The manuscript sounds scientifically and it is appropriately designed. Prevention plan is thought out as goal settings which is person-centred and based on risk-factor monitoring. Risk-factor monitoring is very detailed and practical in the same time (web-based clinician portal, SMS messages, a mobile application (app), and a wearable device) . Secondary prevention goals are set so that they are both expertly guided and that the needs of the patients are listened to (the study participants-patients set them together with the researchers). They provided access and training in technology use. Feasibility outcomes included recruitment, retention, usability, acceptability, and satisfaction. Secondary outcomes included goal attainment, health outcomes, and program costs. Four figures and five tables are sufficiently informative, but also transparent and easy to understand. The conclusions consistent with the evidence and arguments presented. The data and analyses are presented statistically appropriately. The English language appropriate and understandable.

Response: We thank the reviewer for their time in reviewing our paper and their comments.